# TERT Immunohistochemistry in Thin Melanomas Compared to Melanocytic Nevi

**DOI:** 10.3390/diagnostics15091171

**Published:** 2025-05-04

**Authors:** Iulia Zboraș, Loredana Ungureanu, Simona Corina Șenilă, Bobe Petrushev, Paula Zamfir, Doinița Crișan, Flaviu Andrei Zaharie, Ștefan Cristian Vesa, Rodica Cosgarea

**Affiliations:** 1Department of Dermatology, “Iuliu Hațieganu” University of Medicine and Pharmacy, 400006 Cluj-Napoca, Romania; iuliazboras@yahoo.com (I.Z.); simonasenila@yahoo.com (S.C.Ș.); cosgarear@yahoo.com (R.C.); 2Department of Pathology, Regional Institute of Gastroenterology and Hepatology, 400162 Cluj-Napoca, Romania; bobe.petrushev@gmail.com (B.P.); paulacristinela@yahoo.com (P.Z.); 3Department of Pathology, “Iuliu Hațieganu” University of Medicine and Pharmacy, 400006 Cluj-Napoca, Romania; doinitacrisan@gmail.com; 4Faculty of Medicine, “Iuliu Hațieganu” University of Medicine and Pharmacy, 400012 Cluj-Napoca, Romania; zandrei75@yahoo.com; 5Department of Pharmacology, Toxicology and Clinical Pharmacology, “Iuliu Hațieganu” University of Medicine and Pharmacy, 400337 Cluj-Napoca, Romania; stefanvesa@gmail.com

**Keywords:** TERT, expression, telomerase, melanoma, melanocytic nevi

## Abstract

**Background/Objectives**: Telomerase plays a vital role in preserving telomere length, a key process in cancer development. Human telomerase reverse transcriptase (hTERT) is commonly expressed in various cancers, including melanoma. This study evaluated hTERT protein expression in melanomas compared to melanocytic nevi. **Methods**: In total, we examined 75 melanocytic lesions using TERT immunohistochemistry on paraffin-embedded tissues; 36 of them were thin melanomas (Breslow index ≤ 1 mm) and 39 melanocytic nevi. **Results**: The TERT expression differed with statistical significance between the two studied groups, melanomas and melanocytic nevi, in all three aspects examined: percentage of staining (*p* = 0.006), intensity of staining (*p* = 0.035), and localisation of staining (*p* = 0.012). Three quarters of the melanomas stained in over 50% of the cells at cytoplasmic level, 52.78% of the melanomas exhibited an intensity of 3+, and all melanomas were stained at the cytoplasmic level, except for the two negative cases. The values were lower in the melanocytic nevi group. Still, the diagnostic values were relatively low (sensitivity = 75%, specificity = 58.97%, PPV = 62.79%, NPV = 71.88%, and ACC = 66.67%). **Conclusions**: TERT immunohistochemistry differed between the two studied groups; however, the diagnostic utility is low in our study. Combining with other immunohistochemical antibodies would probably increase the diagnostic power.

## 1. Introduction

Telomerase, a ribonucleoprotein reverse transcriptase, facilitates the synthesis of telomeres and helps maintain their length [1]. Telomeres, located at the ends of chromosomes, are made up of repeated “TTAGGG” sequences. They are crucial in preserving genetic information throughout cell division [2]. When telomerase activity is reduced, cells undergo apoptosis, as observed in almost all somatic cells. In contrast, telomerase becomes activated in many cancers, supporting continuous cell proliferation [3]. Telomerase consists of two primary components: the functional RNA component (hTERC), which acts as a template for telomeric DNA synthesis, and the catalytic protein (hTERT), which possesses reverse transcriptase activity [4,5]. The TERT gene, found on chromosome 5, encodes the hTERT catalytic component of the telomerase enzyme [6,7]. In 2013, two somatic mutations were identified in the promoter region of the TERT gene, which are linked to increased TERT expression in melanoma [8,9]. Mutations in the promoter region of the TERT gene tend to arise in the initial stages of melanoma development [10]. They form new sites of binding for E-twenty-six (ETS) and ternary complex transcription factors (TCF), resulting in increased TERT expression [9,11,12]. ETS facilitates NF-κB binding to the mutant TERT promoter, while TRF2 binds to the wild-type TERT promoter. NF-κB and TRF2 activation could impact the TERT gene regulation [13]. The activation or the upregulation of the TERT gene can also happen through other genetic alterations like TERT gene amplifications, TERT gene rearrangements, or epigenetic alterations like TERT promoter hypermethylation [14,15,16,17]. TPP1 promoter mutations work in conjunction with TERT activation to promote telomere maintenance and cellular immortalisation in melanoma [18]. MiR-138-5p targets hTERT and the overexpression of this miRNA showed reduced hTERT protein expression and a reduction in telomerase activity [19].

Telomerase is a target in developing cancer therapies [20]. A telomerase-targeting therapeutic cancer vaccine, UV1, has been developed. This approach has been studied across various cancer types, like melanoma, non-small cell lung cancer, and prostate cancer. In a phase I/IIa clinical trial, the aforementioned vaccine was administered alongside Ipilimumab, an anti-CTLA-4 monoclonal antibody, resulting in a faster and more frequent immune response [21]. Ibrahim et al. observed that adding maintenance doses of the UV1 vaccine every three months in melanoma patients led to a higher tumour shrinkage and a higher probability of immune response [22]. There are also other telomerase-targeted cancer vaccines in clinical trials [23]. Imeltestat, a telomerase inhibitor, received approval in the United States in 2024 for treating myelodysplastic syndromes and is under study in more myeloproliferative disorders and non-small-cell lung cancer [24,25,26,27]. A new therapy targeting telomerase, 6-Thio-2′-deoxyguanosine (6TdG), showed promising results in recent studies on small cell lung cancer in mice and human cell lines [28]. The two telomerase-targeting therapies, Imetelstat and 6TdG, combined with chemotherapy regimes, showed improved survival in telomerase-positive neuroblastoma cell lines [29]. Moreover, BRAF-mutated melanomas with increased TERT expression were observed to be resistant to BRAF and MEK inhibitors. For this reason, the simultaneous inhibition of TERT is crucial for a better response to therapy [30]. Telomerase-targeting therapies could also be promising for the management of acral melanomas [31].

The TERT expression can be detected and measured by immunohistochemistry (IHC) as TERT protein levels in tissue samples, by real-time polymerase chain reaction (RT-PCR) or by in situ hybridisation (ISH) as TERT mRNA levels, by Western blot as protein levels in cell or tissue lysates, or by PCR-based telomeric repeat amplification protocol (TRAP) assays as activity levels of the telomerase [12,14,32,33].

The immunohistochemistry method is simple, cost-effective, fast, and available in almost all histopathological laboratories [34]. The results in the previously published studies are variable regarding the localisation of the staining, the pattern of the staining, or the diagnostic and prognostic utility [32,35,36,37,38]. The positivity of TERT expression did not correlate with TERT promoter mutations [32,36,37].

Our study evaluated whether there are statistically significant differences between TERT immunohistochemistry expression in melanocytic nevi and thin melanomas regarding the percentage, intensity, and localisation of the staining or whether it is associated with other clinicopathological variables. We tried to evaluate the possible diagnostic power in differentiating between benign melanocytic lesions and malignant melanocytic lesions. Only a few studies have been published on TERT immunohistochemistry in melanoma, and none have focused on thin melanomas. No published study included a Romanian population.

## 2. Materials and Methods

### 2.1. Research Design and Methodology

We conducted a retrospective study involving 36 thin melanomas and 39 melanocytic nevi diagnosed at the Department of Dermatovenerology of the Cluj-Napoca Emergency County Hospital from 2014 to 2018. The study was approved by the Ethics Committee of the Iuliu Hațieganu University of Medicine and Pharmacy Cluj-Napoca, Romania (protocol code 39 from 31 March 2023). All participants provided informed consent, and the study was carried out under the Declaration of Helsinki.

### 2.2. Immunohistochemical Analysis

Paraffin-embedded tissue blocks were obtained from the Department of Pathology at Cluj-Napoca Emergency County Hospital and were reviewed for diagnostic confirmation by a pathologist (D.C.). We subsequently prepared 5 mm thick tissue sections for immunohistochemical analysis and carried out depigmentation using 3% hydrogen peroxide to eliminate excess melanin. Afterwards, we performed the immunohistochemical staining using the Anti-Telomerase Catalytic Subunit antibody—600-401-252 (rabbit) produced by Rockland in Limerick, Pennsylvania, USA, as they used in the study by Hugdahl et al. [37]. The reliability of the antibody was previously validated in the study by Wu et al. [39]. The staining was carried out on the automated stainer platform from Leica Biosystems, Leica Bond-Max, which was manufactured in Melbourne, UK. We performed the staining using a ph 6 buffer at a 1:500 dilution with a DAB brown chromogen. For the negative control, we omitted the antibody. Eccrine glands and ducts served as positive internal controls. The cases were interpreted by an experienced board-certified anatomopathologist (B.P.), blinded to the patient’s characteristics.

Staining results were documented based on the percentage, intensity, and localisation of the immunoreactive cells. The percentage rate was grouped into 0%, <10%, 10–50%, and >50%. Colour intensity was classified as negative (0), weak (1+), moderate (2+), or strong (3+). Staining localisation was recorded as cytoplasmic, nuclear, or nuclear and cytoplasmic, similar to other studies using TERT antibodies for immunohistochemistry on malignant tumours [40,41,42]. Due to its canonical functions, telomerase is mainly expressed at the nuclear level, but due to its non-canonical functions, it can also be expressed at the cytoplasmic level [43,44].

### 2.3. Data Analysis and Statistics

All statistical analyses were conducted using MedCalc^®^ Statistical Software version 23.1.6 from MedCalc Software Ltd., Ostend, Belgium (https://www.medcalc.org; 2025). Continuous variables (e.g., age, Breslow index, mitotic rate) were reported as a median and interquartile range since their distribution was not normal. Categorical variables were presented as frequencies and percentages. The Chi-square test (or Fisher’s exact test when expected frequencies were below five) assessed differences between categorical variables, such as TERT staining percentage, intensity, and localisation between melanomas and melanocytic nevi. Given their non-normal distribution, the Mann–Whitney U test was applied to compare continuous variables (e.g., age, Breslow index, mitotic rate) across the study groups. A *p*-value under 0.05 was regarded as statistically significant.

## 3. Results

Using immunohistochemistry, we examined 36 cases of thin melanomas and 39 cases of melanocytic nevi for TERT protein expression (Figure 1). The melanoma group included 10 stage 0 melanomas, 14 stage IA melanomas, and 12 stage IB melanomas. The majority, 31 melanomas, were superficial spreading melanomas (SSM), 1 lentigo maligna (LM), 2 lentigo maligna melanomas (LMM), and 2 acral lentiginous melanomas (ALM). Patient ages ranged from 26 to 85 years; the mean age was 59.47 and the median was 62. There were 14 female patients and 22 male patients. Regarding the localisation of the tumour, 4 melanomas were located in the head and neck region, 19 in the trunk region, 6 in the upper limb region, and 7 in the lower limb region. A mean Breslow index of 0.45 mm and a median Breslow index of 0.5 mm were observed. The mean mitotic rate was 1.52/mm^2^, and the median mitotic rate was 1/mm^2^. In total, 13 melanomas were in the vertical growth phase, while 23 were in the horizontal growth phase. Ten melanomas had developed on a preexistent nevus. A personal history of melanoma was reported in two patients, whereas no patients had a documented family history of melanoma.

The melanocytic nevi group included 31 low-grade dysplastic nevi, 1 high-grade dysplastic nevus, 5 dysplastic nevi with an unspecified grade of dysplasia, 1 Halo nevus (Sutton), and 1 common melanocytic nevus (dermal nevus). In total, 27 melanocytic nevi were compound, 11 junctional nevi, and 1 intradermal nevus. Regarding the localisation, 1 was located in the head and neck region, 26 on the trunk, 4 on the upper limb, and 8 on the lower limb. Two of the low-grade dysplastic nevi had acral localisation. Patient ages ranged from 16 to 67 years, with a mean age of 35.23 and a median age of 33. There were 21 female patients and 18 male patients. No personal history of melanoma was reported, but six patients had a family history of the condition.

About 75% of the melanomas stained in over 50% of the cells, while only 41% of the nevi stained in over 50%. If the cutoff of positivity is 50%, three-quarters of the melanomas were positive for TERT expression, while one-quarter were negative. The positivity in the melanocytic nevi group was lower (41%) than in the melanoma group (75%) (*p* = 0.006). Regarding the intensity of the staining, more than half of the melanomas were stained with an intensity of 3+, while under one-third of the nevi were stained with this intensity (*p* = 0.035). All melanomas were stained at the cytoplasmic level except for the two negative cases. In contrast, some melanocytic nevi were stained at the nuclear level or both nuclear and cytoplasmic levels (*p* = 0.012) (Table 1). For a hypothetical cutoff value of more than 50% stained cells being considered a positive case, the sensitivity was 75%, the specificity 58.97%, the positive predictive value (PPV) 62.79%, the negative predictive value (NPV) 71.88% and the accuracy (ACC) 66.67%.

Variations in TERT expression between benign and malignant melanocytic lesions could have significant clinical relevance, especially in supporting early cancer diagnosis. Integrating TERT staining into pathology assessments could enhance diagnostic accuracy.

The TERT staining in melanoma was present in a higher percentage of the cells, with higher intensity, and only at the cytoplasmic level compared to melanocytic nevi (Figure 2). Most melanomas stained in over 50% of the cells with an intensity of 3+. We used the eccrine glands/ducts as positive internal controls (Figure 3).

When comparing the two melanoma groups—melanoma stage 0 and melanoma stage I—we observed no statistically significant difference regarding the staining percentage, intensity, or localisation of the staining (Table 2). However, there was a trend towards significance regarding the staining percentage (*p* = 0.09). The cases that stained in over 50% of the cells were more numerous in the group of melanomas stage I compared to melanomas stage 0 (in situ). All melanomas were stained only at the cytoplasmic level.

When we compared the patients with melanomas positive for the TERT staining in over 50% of the cells and the patients with melanomas that stained in under 50% according to clinicopathological variables, a significant difference was observed between the two groups regarding age (*p* = 0.031), the personal history of melanoma (*p* = 0.012), and melanomas developed on a preexistent nevus (*p* = 0.032) (Table 3). The melanoma patients whose tumours stained positive for TERT in over 50% of the cells were older (median = 64 years) than those with melanomas that stained in under 50% (median = 51 years).

No patient with melanoma that stained in over 50% of the cells reported a personal history of melanoma. In comparison, two patients with melanomas that stained in under 50% of the cells reported a personal history of melanoma. The first patient had two other melanomas diagnosed, and the second had five other melanomas diagnosed. No patient included in the study had a known family history of melanoma. A third of the melanomas that stained in over 50% of the cells developed on a preexisting nevus, while none that stained in under 50% of the cells developed on a preexisting nevus. There was also a tendency for significant variations between the two groups regarding the histopathological stage (*p* = 0.074) and the mitotic rate (*p* = 0.079). The majority of the melanomas stage IA and IB stained in over 50% of the cells, while half of the melanomas in situ stained in over 50% of the cells and half of them stained in under 50% of the cells. The median mitotic rate was higher in the group of melanomas which showed staining in over 50% of the cells compared to the group with staining in under 50%.

We compared the three different groups of melanomas according to the staining percentage: over 50% of the cells, 10–50% of the cells, or entirely negative. A significant difference was still observed in relation to age (*p* = 0.034) and personal history of melanoma (*p* = 0.012) (Table 4). No melanoma stained in 1–9% of the cells. The median age was higher in the group of melanomas that stained in over 50% of the cells (median age = 64 years) compared to the group that stained in 10–50% of the cells (median age = 58) and compared to the entirely negative cases (median age = 47 years). No patient with melanoma that stained in over 50% of the cells reported a personal history of melanoma. Both patients who reported a personal history of melanoma had melanomas that stained in 10–50% of the cells.

When comparing melanocytic nevi with staining in over 50% of cells to those with staining in under 50%, we found no significant difference between the two groups according to clinicopathological variables like sex, age, histopathological subtype, melanocyte localisation, or nevus localisation (Table 5).

## 4. Discussion

A few studies have been published on TERT immunohistochemistry in melanocytic lesions with highly variable results. Most used the same antibody as in our research at the same or different dilutions, but some used a different antibody. Studies that used the same antibody had more similar results. For example, in the survey conducted by Hugdahl et al., the TERT staining in melanomas was described as homogenous and localised at the cytoplasmic level, as in our study. They used the antibody we used in this study at a different dilution (1:125). The results were interpreted using a staining index, calculated by multiplying the intensity and proportion scores. TERT protein expression was linked to greater tumour thickness and decreased survival [37]. Bustos et al. observed TERT expression in all melanocytic lesions, but the pattern was homogenous in melanocytic nevi and homogeneous or heterogeneous in melanomas. The heterogeneous pattern was associated with a higher Breslow index, fast-growing melanomas, higher mitotic rate, and TERT promoter mutations [38]. In the study conducted by Populo et al., TERT protein expression in melanomas was nuclear and cytoplasmic. We observed nuclear and cytoplasmic expression only in melanocytic nevi. They used the same antibody we used in this study at the same dilution (1:500) and observed no correlation with TERT promoter mutational status, disease-free survival, or overall survival [36]. In this study, there was no association with tumour thickness. However, we observed a trend toward statistical significance regarding the histopathological stage and the mitotic rate. The number of included patients was low, but probably in a larger cohort, the results would have been statistically significant. We found a significant association between TERT protein expression and the patient’s age, the melanoma’s development on a preexisting nevus and the absence of a prior personal diagnosis of melanoma. The association with age or the personal history of melanoma could be spurious because the cohort was small.

Cytoplasmic staining was also observed in other malignant tumours like hepatocellular carcinoma, lung cancer, and cervical cancer [40,41,42]. The signification of cytoplasmic TERT staining associated more frequently with malignant tumours could be due to the non-canonical functions of telomerase reverse transcriptase, like support of cell growth and proliferation, cell cycle progression, and maintenance of mitochondrial integrity in response to the oxidative stress [43,44]. The canonical function of the telomerase, localised at the nuclear level, is to keep the length of the telomeres and enable the cell to achieve unlimited proliferation potential [45]. In the case of oxidative stress telomerase can migrate from nuclear level to cytoplasmic/mitochondrial level [46].

Studies that used other antibodies had different results. For example, Fullen et al. used the hTERT antibody from Novocastra, Newcastle, UK. They observed no statistically significant difference between TERT expression in melanocytic nevi and melanomas, so they considered that TERT immunohistochemistry has no diagnostic role in differentiating benign from malignant melanocytic lesions. Still, they observed an increase in mean TERT expression from acquired nevi to dysplastic nevi and melanomas. The number of included cases was low and they failed to prove a statistically significant difference [35]. In the study by Kohli et al., TERT immunohistochemistry was noted to be localised more frequently at the nucleolus level in benign nevi while at the “non-nucleolar” level in melanomas. They used a different antibody, TERT (EST21-A) from Alpha Diagnostic International at a dilution of 1:50 [32].

In the study conducted by Cho et al. on acral melanoma, TERT protein expression by immunohistochemistry was present in more than half of acral lentiginous melanomas and nonacral cutaneous melanoma, in 40% of the nonlentiginous acral melanomas, in all metastatic acral lentiginous melanomas and no acral nevi. TERT protein expression was not linked with reduced disease-specific survival, although there was a trend toward reduced overall survival without statistical significance [47]. A recent study showed TERT protein expression in half of the primary acral lentiginous melanomas and all metastatic melanomas with variable percentages and intensities. Similar to our research, TERT expression was mainly cytoplasmic. They used an anti-TERT antibody from Abcam, Cambridge, Massachusetts (clone Y182) at a dilution of 1:100. The sensitivity for predicting TERT gene amplification through TERT protein expression by immunohistochemistry was 100%, while the specificity was only 57%; thus, the clinical utility is low [48]. In acral lentiginous melanoma, TERT amplification was observed to be involved in the progression to metastatic disease [49]. Our study included two cases of acral lentiginous melanoma and two cases of low-grade dysplastic nevi with acral localisation. The first melanoma case was stage IA melanoma, positive in over 50% of the cells with an intensity of 3+. The second was a melanoma in situ, positive in 10–50% of the cells with an intensity of 2+. In contrast, one dysplastic nevus with acral localisation was positive in 10–50% of the cells with an intensity of 2, while the other was negative.

Other methods that detect TERT expression are PCR-based telomeric repeat amplification protocol (TRAP), in situ hybridisation (ISH), real-time polymerase chain reaction (RT-PCR), and Western blot [12,14,32,33]. PCR-based TRAP is the method that detects and quantifies telomerase activity. Higher levels of telomerase activity were noted in melanoma metastases and primary melanomas, and lower levels in dysplastic nevi followed by benign melanocytic nevi [33,50,51,52]. Rudolph et al. observed that the telomerase activity level can have a prognostic value. Early metastasis occurred in cases with high telomerase activity, while cases with lower telomerase activity were linked to absent or slow progression of the disease [52]. In the study by Tosi et al., Spitz nevi showed a lower telomerase activity compared to melanomas [53]. Guttman-Yasky et al. determined hTER expression by in situ hybridisation (ISH), which is less sensitive than the TRAP method. They observed no statistically significant difference among Spitz nevi, common melanocytic nevi and melanomas [54].

Melanomas harbouring TERT promoter mutations are associated with elevated TERT expression by RT-PCR, expressed as high TERT mRNA levels compared to melanomas without mutations [55,56]. TERT promoter mutations in melanomas were linked to decreased disease-specific and overall survival [57,58,59]. TERT promoter methylation is also linked with increased TERT expression, as demonstrated by TERT mRNA in situ hybridisation (ISH) [14]. Increased TERT expression was observed in metastases from primary melanomas with a Breslow index under 2 mm and could contribute to early metastasis [60]. TERT promoter methylation, combined with TERT promoter mutations or without showed reduced recurrence-free survival in adolescent and young adult melanoma patients [61].

There is a lot of variability between the different studies published on TERT immunohistochemistry in melanocytic lesions. Different antibodies could explain some differences, but not all of them. This study was conducted on a limited cohort of Romanian melanoma patients and focused exclusively on thin melanomas. No previous research has focused on thin melanomas, which is the novelty of this study. There are also possible genetic, clinical, and histopathological differences in the Romanian population compared to other Eastern populations. The reported BRAF mutation in Romanian melanoma patients is similar to that in other countries [62,63]. The clinical and histopathological characteristics are also comparable to those of other European countries. There is a predominance of melanomas in female patients, similar to that of the Spanish and French populations. The median age at diagnosis is also close to that in other European countries. Superficial spreading melanoma is the most frequent melanoma subtype, similar to Spain and Serbia [64].

A statistically significant difference was observed between melanomas and melanocytic nevi regarding percentage, intensity, and localisation of the colouration. However, if we considered a hypothetical cutoff value of the presence of the staining in more than 50% of the cells as being positive, the specificity (58.97%) and accuracy (66.67%) were low. The diagnostic value and, consequently, clinical utility of the TERT immunohistochemistry on its own are relatively limited; probably, combined with other immunohistochemical stainings, the diagnostic value would improve. However, if we considered positive cases only, the ones that stained in over 50% of the cells at the cytoplasmic level, the diagnostic value would improve slightly. Still, correlations with other immunohistochemical stainings would be beneficial. 

Further studies on this method, including larger cohorts and correlations with other methods detecting TERT expression, could bring valuable information. Hugdahl et al. observed a correlation between TERT expression and reduced survival, meaning that this immunohistochemical colouration could have a prognostic role in more advanced cases [37].

## 5. Conclusions

The TERT immunohistochemistry differed with statistical significance between the two studied groups in all three aspects: percentage of the staining, intensity of the staining, and localisation of the staining. Melanomas were stained only at the cytoplasmic level, and the percentage and intensity of the staining were greater in the melanoma group than in the melanocytic nevi group. However, the diagnostic power was relatively low in our study. Still, the combination with other immunohistochemical antibodies could increase the diagnostic utility. Larger multicenter studies on larger cohorts are needed. This immunohistochemical analysis may provide valuable support to conventional diagnostic criteria, but it is not intended to replace them.

## Figures and Tables

**Figure 1 diagnostics-15-01171-f001:**
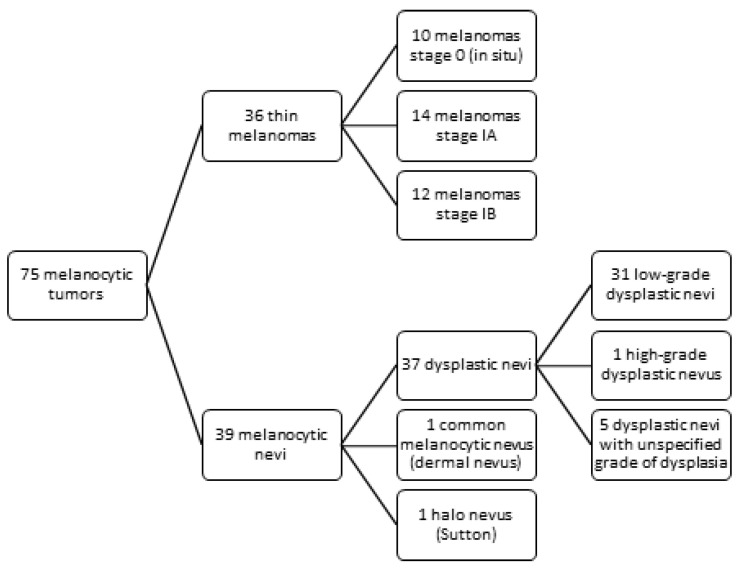
Study design—melanocytic tumors distribution.

**Figure 2 diagnostics-15-01171-f002:**
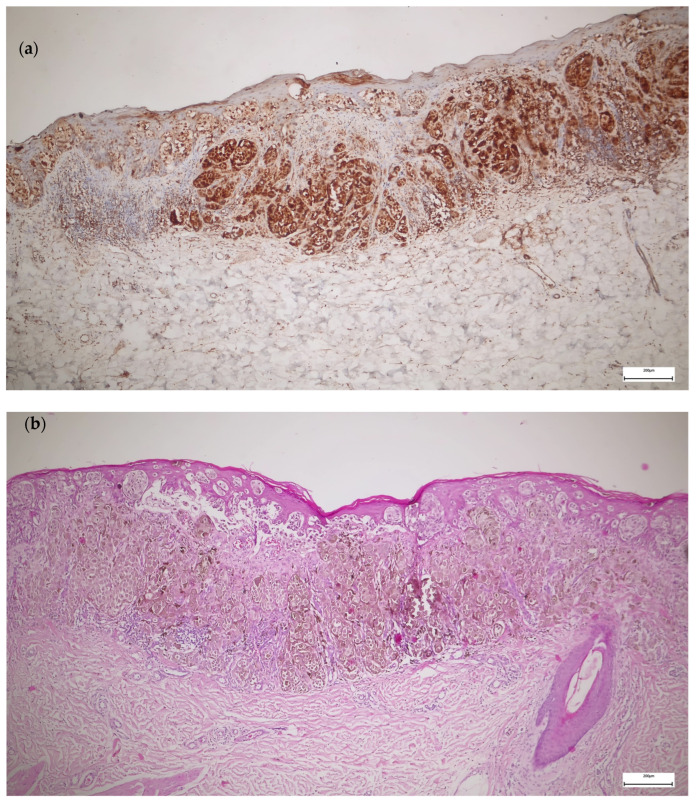
TERT versus hematoxylin and eosin (H&E) immunohistochemistry in melanoma: (**a**) TERT IHC—superficial spreading melanoma stage IB, staining in more than 50% of the cells, intensity 3+, cytoplasmic staining; (**b**) H&E IHC of the same case—superficial spreading melanoma stage IB.

**Figure 3 diagnostics-15-01171-f003:**
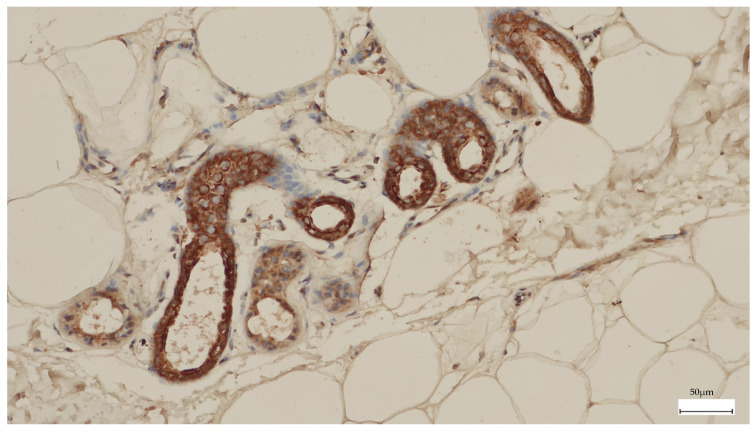
The eccrine glands and ducts were used as a positive internal control for TERT immunohistochemistry.

**Table 1 diagnostics-15-01171-t001:** TERT immunohistochemical analysis in melanomas with Breslow index ≤ 1 mm compared to melanocytic nevi.

Variable		Nevus	Melanoma	Chi-Square *p*-Value
**Staining percentage N (%)**	**0%**	8 (20.51)	2 (5.56)	0.006
	**<10%**	6 (15.38)	0 (0)	
	**10–50%**	9 (23.08)	7 (19.44)	
	**>50%**	16 (41.03)	27 (75)	
**Color intensity N (%)**	**0**	8 (20.51)	2 (5.56)	0.035
	**1+**	9 (23.08)	5 (13.89)	
	**2+**	13 (33.33)	10 (27.78)	
	**3+**	9 (23.08)	19 (52.78)	
**Staining localisation N (%)**	**Negative**	8 (20.51)	2 (5.56)	0.012
	**Cytoplasmic**	25 (64.1)	34 (94.44)	
	**Nuclear**	3 (7.69)	0 (0)	
	**Nuclear and cytoplasmic**	3 (7.69)	0 (0)	

**Table 2 diagnostics-15-01171-t002:** TERT immunohistochemistry in melanoma in situ versus melanoma stage I.

Variable		Melanoma Stage 0 (In Situ)	Melanoma Stage I	*p*-Value
**Staining percentage N (%)**	**0%**	1 (10)	1 (3.85%)	0.09
	**<10%**	-	-	
	**10–50%**	4 (40)	3 (11.53%)	
	**>50%**	5 (50)	22 (84.62%)	
**Color intensity N (%)**	**0**	1 (10)	1 (3.85%)	0.4
	**1+**	2 (20)	3 (11.54%)	
	**2+**	1 (10)	9 (34.61%)	
	**3+**	6 (60)	13 (50%)	
**Staining localisation N (%)**	**Negative**	1 (10)	1 (4%)	0.4
	**Cytoplasmic**	9 (90)	25 (96%)	

**Table 3 diagnostics-15-01171-t003:** Comparison of TERT staining in melanomas with over 50% positive cells versus those with under 50% positive cells, based on clinicopathological variables.

Variable		TERT Negative Melanoma (10–50%, <10%, 0%)	TERT Positive Melanoma (>50%)	*p*-Value
**Sex N (%)**	**M**	4 (44.44)	18 (66.67)	0.236
	**F**	5 (55.56)	9 (33.33)	
**Median age [years] [IQR]**		51 (49; 58)	64 (59; 68.5)	0.031
**Localisation N (%)**	**Head and neck**	1 (11.11)	3 (11.11)	0.460
	**Trunk**	4 (44.44)	15 (55.56)	
	**Upper limb**	3 (33.33)	3 (11.11)	
	**Lower limb**	1 (11.11)	6 (22.22)	
**Histopathological subtype N (%)**	**SSM**	8 (88.89)	23 (85.19)	0.642
	**LMM**	0 (0)	2 (7.41)	
	**LM**	0 (0)	1 (3.7)	
	**ALM**	1 (11.11)	1 (3.7)	
**Vertical growth N (%)**	**No**	6 (66.67)	17 (62.96)	0.841
	**Yes**	3 (33.33)	10 (37.04)	
**Stage N (%)**	**In situ**	5 (55.56)	5 (18.52)	0.074
	**IA**	3 (33.33)	11 (40.74)	
	**IB**	1 (11.11)	11 (40.74)	
**Ulceration N (%)**	**No**	9 (100)	26 (96.3)	0.558
	**Yes**	0 (0)	1 (3.7)	
**Regression N (%)**	**No**	8 (88.89)	22 (81.48)	0.606
	**Yes**	1 (11.11)	5 (18.52)	
**Median Breslow index [IQR]**		0 (0; 0.75)	0.6 (0.375; 0.7)	0.193
**Median mitotic rate [IQR]**		0 (0; 1)	1 (0; 3)	0.079
**Personal history of melanoma**	**No**	7 (77.78)	27 (100)	0.012
	**Yes**	2 (22.22)	0 (0)	
**Nevus-associated melanoma**	**No**	9 (100)	17 (62.96)	0.032
	**Yes**	0 (0)	10 (37.04)	

SSM—superficial spreading melanoma, LMM—lentigo maligna melanoma, LM—lentigo maligna, ALM—acral lentiginous melanoma.

**Table 4 diagnostics-15-01171-t004:** Comparison between TERT staining percentages in patients with melanoma according to clinicopathological variables.

Variable		>50%	10–50%	0%	*p*-Value
**Sex N (%)**	**M**	18 (66.67)	3 (42.86)	1 (50)	0.488
	**F**	9 (33.33)	4 (57.14)	1 (50)	
**Median age [years] [IQR]**		64 (59; 68.5)	58 (50.5; 59.5)	47 (46; 48)	0.034
**Localisation N (%)**	**Head and neck**	3 (11.11)	1 (14.29)	0 (0)	0.747
	**Trunk**	15 (55.56)	3 (42.86)	1 (50)	
	**Upper limb**	3 (11.11)	2 (28.57)	1 (50)	
	**Lower limb**	6 (22.22)	1 (14.29)	0 (0)	
**Histopathological subtype N (%)**	**SSM**	23 (85.19)	6 (85.71)	2 (100)	0.892
	**LMM**	2 (7.41)	0 (0)	0 (0)	
	**LM**	1 (3.7)	0 (0)	0 (0)	
	**ALM**	1 (3.7)	1 (14.29)	0 (0)	
**Vertical growth N (%)**	**No**	17 (62.96)	5 (71.43)	1 (50)	0.840
	**Yes**	10 (37.04)	2 (28.57)	1 (50)	
**Stage N (%)**	**In situ**	5 (18.52)	4 (57.14)	1 (50)	0.239
	**IA**	11 (40.74)	2 (28.57)	1 (50)	
	**IB**	11 (40.74)	1 (14.29)	0 (0)	
**Ulceration N (%)**	**No**	26 (96.3)	7 (100)	2 (100)	0.842
	**Yes**	1 (3.7)	0 (0)	0 (0)	
**Regression N (%)**	**No**	22 (81.48)	6 (85.71)	2 (100)	0.781
	**Yes**	5 (18.52)	1 (14.29)	0 (0)	
**Median Breslow index [IQR]**		1 (0; 3)	1 (0; 1)	0 (0; 0)	0.336
**Median mitotic rate [IQR]**		0.6 (0.38; 0.7)	0 (0; 0.72)	0.45 (0.22; 0.68)	0.632
**Personal history of melanoma**	**No**	27 (100)	5 (71.43)	2 (100)	0.012
	**Yes**	0 (0)	2 (28.57)	0 (0)	
**Nevus-associated melanoma**	**No**	17 (62.96)	7 (100)	2 (100)	0.099
	**Yes**	10 (37.04)	0 (0)	0 (0)	

**Table 5 diagnostics-15-01171-t005:** Comparison between melanocytic nevi according to the percentage of the staining and clinicopathological variables.

Variable		TERT Negative Nevus (10–50%, <10%, 0%)	TERT Positive Nevus (>50%)	*p*-Value
**Sex N (%)**	**M**	11 (47.83)	7 (43.75)	0.802
	**F**	12 (52.17)	9 (56.25)	
**Median age (years) [IQR]**		34 (27.5; 44)	25.5 (21.75; 41.5)	0.803
**Histopathological subtype N (%)**	**Low-grade dysplastic nevus**	20 (86.96)	11 (68.75)	0.319
	**High-grade dysplastic nevus**	0 (0)	1 (6.25)	
	**Dysplastic nevus with an unspecified grade of dysplasia**	2 (8.7)	3 (18.75)	
	**Halo nevus**	0 (0)	1 (6.25)	
	**Common melanocytic nevus (Dermal nevus)**	1 (4.35)	0 (0)	
**Melanocytes localisation N (%)**	**Compound nevus**	14 (60.87)	13 (81.25)	0.346
	**Junctional nevus**	8 (34.78)	3 (18.75)	
	**Intradermal nevus**	1 (4.35)	0 (0)	
**Nevus localisation N (%)**	**Head and neck**	1 (4.35)	0 (0)	0.695
	**Trunk**	14 (60.87)	12 (75)	
	**Upper limb**	3 (13.04)	1 (6.25)	
	**Lower limb**	5 (21.74)	3 (18.75)	

## Data Availability

Data used or analysed during this study can be obtained from the corresponding author upon reasonable request.

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
