# Peer review of "TERT Immunohistochemistry in Thin Melanomas Compared to Melanocytic Nevi"

_diagnostics, 2025, doi:10.3390/diagnostics15091171_

Round 1
Reviewer 1 Report
Comments and Suggestions for Authors
I have read this manuscript with pure interest. Personally I agree that the topic is relevant and timely as it focused on diagnostic and prognostic biomarkers for melanoma.
- While the study design is relatively straightforward and the analysis includes some interesting observations, I have some major comments:
- The authors focus exclusively on IHC without integrating molecular assays like TERT promoter mutation analysis (e.g., by PCR or sequencing), which are known to correlate with TERT expression. This limits the mechanistic understanding and weakens the conclusions. The authors acknowledge this as a limitation but do not adequately emphasize its impact.
- The sample sizes for both melanomas (n=36) and nevi (n=39) are relatively small. Subgroup analyses (e.g., melanoma in situ vs. stage IA/IB) further fragment the data, risking type II errors. Power analysis is not discussed. For example, the comparison of TERT positivity across melanoma stages (Table 2) shows trends but lacks significance—likely due to underpowering.
- The use of a >50% threshold for “positive” staining is arbitrary. There is no justification or ROC analysis to support this cut-off as diagnostically or clinically meaningful. The reported diagnostic metrics (e.g., 66.67% accuracy) suggest limited clinical utility, which is not critically discussed.
- The classification of nevi includes "dysplastic nevi with unspecified grade" and "common melanocytic nevus." These categories are too vague and potentially heterogeneous. Precise histological criteria should be stated.
Minor comments:
The manuscript reports on cytoplasmic vs. nuclear localization of TERT staining but does not explore the biological implications or validate findings with dual localization markers. I strongly believe that further discussion is required.
To reexamine the analysis: The association between TERT staining and variables such as age or personal history of melanoma (Table 3 and Table 4) is likely spurious due to small numbers. For instance, the "significant" lack of personal melanoma history in TERT-positive cases (p=0.012) is based on just two cases.
Abstract: The abstract is too vague in stating "statistically significant" differences without actual values or context. Include key statistics (e.g., p-values, sensitivity, specificity).
The antibody dilution (1:500) and platform (Leica Bond-Max) are described, but the validation of antibody specificity and reproducibility is not discussed. Was inter-observer variability assessed?
Figures (particularly Figure 2) are dense and difficult to interpret without clearer labeling. Consider simplifying and focusing on representative examples with higher image quality.
Numerous typographic issues exist throughout the manuscript, such as "cytoplasmatic" instead of "cytoplasmic." Please revise the manuscript for grammar and consistency.
Author Response
1. Summary |
|
Thank you very much for your answer and the opportunity to improve the manuscript. Please find the detailed response below and the corresponding corrections highlighted in the resubmitted file.
|
|
2. Point-by-point response to Comments and Suggestions for Authors
|
|
Comments 1: The authors focus exclusively on IHC without integrating molecular assays like TERT promoter mutation analysis (e.g., by PCR or sequencing), which are known to correlate with TERT expression. This limits the mechanistic understanding and weakens the conclusions. The authors acknowledge this as a limitation but do not adequately emphasize its impact. |
|
Response 1: Thank you for highlighting this point. We agree with your observation that a more detailed analysis of TERT promoter mutations could have been included. We will take this into consideration for our future research. However, as noted in the literature, the correlation between TERT promoter mutations and TERT expression is generally low [see page 3, paragraph 1, lines 92-93]. Additionally, immunohistochemistry is a rapid and widely accessible method in histopathological laboratories, and may have greater clinical relevance. Nevertheless, integrating other techniques to assess TERT expression could have provided further valuable insights [page 12, paragraph 5, lines 391-392].
|
|
Comments 2: The sample sizes for both melanomas (n=36) and nevi (n=39) are relatively small. Subgroup analyses (e.g., melanoma in situ vs. stage IA/IB) further fragment the data, risking type II errors. Power analysis is not discussed. For example, the comparison of TERT positivity across melanoma stages (Table 2) shows trends but lacks significance—likely due to underpowering. |
|
Response 2: Thank you for your comment. We agree with your point. For this reason, we chose to compare melanoma in situ with stage I melanomas to avoid excessive fragmentation of the data. Table 2 has been revised accordingly.
Comments 3: The use of a >50% threshold for “positive” staining is arbitrary. There is no justification or ROC analysis to support this cut-off as diagnostically or clinically meaningful. The reported diagnostic metrics (e.g., 66.67% accuracy) suggest limited clinical utility, which is not critically discussed. Response 3: We agree with your comment. The cutoff was chosen hypothetically to explore the potential diagnostic utility of TERT immunohistochemistry. As noted, the accuracy, sensitivity, and specificity are indeed low. However, when combined with other immunohistochemical markers, TERT staining might offer greater diagnostic value [page 12, paragraph 4, lines 385-387].
Comments 4: The classification of nevi includes "dysplastic nevi with unspecified grade" and "common melanocytic nevus." These categories are too vague and potentially heterogeneous. Precise histological criteria should be stated. Response 4: We agree with your comment. We used the term “dysplastic nevi with unspecified grade of dysplasia” to refer to melanocytic nevi that were classified as dysplastic, but for which no specific grade of dysplasia was determined. In contrast, a common melanocytic nevus refers to a dermal nevus, without dysplasia.
Comments 5: The manuscript reports on cytoplasmic vs. nuclear localization of TERT staining but does not explore the biological implications or validate findings with dual localization markers. I strongly believe that further discussion is required. Response 5: Thank you for your comment. We agree with your observation. The TERT IHC staining pattern can vary depending on the tumour type or antibody clone. Hugdahl et al., who used the same antibody as we did, reported cytoplasmic staining in melanomas. Although telomerase is typically localized in the nucleus, in our study, nuclear or combined nuclear and cytoplasmic staining was observed only in melanocytic nevi. It is likely that nuclear staining is characteristic of benign lesions, while cytoplasmic staining is indicative of pathology. Additionally, cytoplasmic staining may reflect the non-canonical functions of telomerase, which are associated with its expression in the cytoplasm and mitochondria. We have expanded on these points in the Discussion section [page 11, paragraph 2, lines 306-314].
Comments 6: To reexamine the analysis: The association between TERT staining and variables such as age or personal history of melanoma (Table 3 and Table 4) is likely spurious due to small numbers. For instance, the "significant" lack of personal melanoma history in TERT-positive cases (p=0.012) is based on just two cases. Response 6: Thank you again for your comment. We believe you are correct, and we have added a few remarks addressing this point [page 12, paragraph 1, lines 303-305].
Comments 7: Abstract: The abstract is too vague in stating "statistically significant" differences without actual values or context. Include key statistics (e.g., p-values, sensitivity, specificity). Response 7: We completely agree with your comment and have added the relevant information to the Abstract section accordingly.
Comments 8: The antibody dilution (1:500) and platform (Leica Bond-Max) are described, but the validation of antibody specificity and reproducibility is not discussed. Was inter-observer variability assessed? Response 8: Thank you for pointing this out. The reliability of the TERT antibody from Rockland was validated in a previous study (Wu et al., 2006, PMID: 16772337), and we have updated the manuscript accordingly [page 3, paragraph 4, lines 117-118]. Inter-observer variability was not assessed, as all cases were interpreted by an experienced, board-certified anatomopathologist (B.P.).
Comments 9: Figures (particularly Figure 2) are dense and difficult to interpret without clearer labeling. Consider simplifying and focusing on representative examples with higher image quality. Response 9: We agree with your comment and have simplified the Figure 2 to highlight a representative case. Thank you!
Comments 10: Numerous typographic issues exist throughout the manuscript, such as "cytoplasmatic" instead of "cytoplasmic." Please revise the manuscript for grammar and consistency. Response 10: Thank you again for your comment. We have carefully revised the manuscript for grammatical accuracy and consistency.
|
Reviewer 2 Report
Comments and Suggestions for Authors
The study lacks novelty, justification for carrying out study on Romanian population, validity and specifity of antibody used, lack of other molecular data such as qPCR, WB and interobserver validity of TERT marker.
The manuscript titled 'TERT Immunohistochemistry in Thin Melanomas Compared to Melanocytic Nevi' presents several major concerns that significantly limit its suitability for publication in this journal:
* The study largely replicates previously published findings on TERT expression in melanomas and nevi, thus offering limited novelty or contribution to the field.
* The manuscript fails to provide a compelling rationale for studying a Romanian population, with no clear justification for potential genetic, epidemiological, or clinical differences that warrant this specific study.
* The authors do not adequately address the well-known challenges related to antibody specificity, staining protocols, and interpretation variability in TERT immunohistochemistry, raising significant concerns about the reproducibility and reliability of their results.
* The limited sample size, particularly within the specific Romanian population studied, compromises the statistical power and generalizability of the findings.
* The study's findings could have been significantly strengthened by the inclusion of molecular data, such as RT-PCR for TERT mRNA expression and/or sequencing for TERT promoter mutations, to validate the immunohistochemical results.
* The study's reliability is further called into question by the absence of an evaluation of interobserver variability in the interpretation of the immunohistochemistry results, a crucial factor in such studies.
Author Response
1. Summary |
|
|
Thank you very much for taking the time to review this manuscript. We truly appreciate the opportunity to improve it. Below, you will find the detailed responses along with the corresponding revisions highlighted in the resubmitted files. We hope that the new revised form is much better than the previous one and will be considered for publication.
|
||
2. Questions for General Evaluation |
Reviewer’s Evaluation |
Response and Revisions |
Does the introduction provide sufficient background and include all relevant references? |
Yes/Can be improved/Must be improved/Not applicable |
We have made the recommended revisions where possible. Please follow the step-by-step responses in the underlying section. |
Are all the cited references relevant to the research? |
Yes/Can be improved/Must be improved/Not applicable |
|
Is the research design appropriate? |
Yes/Can be improved/Must be improved/Not applicable |
|
Are the methods adequately described? |
Yes/Can be improved/Must be improved/Not applicable |
|
Are the results clearly presented? |
Yes/Can be improved/Must be improved/Not applicable |
|
Are the conclusions supported by the results? |
Yes/Can be improved/Must be improved/Not applicable
|
|
3. Point-by-point response to Comments and Suggestions for Authors
|
||
Comments 1: The study largely replicates previously published findings on TERT expression in melanomas and nevi, thus offering limited novelty or contribution to the field. |
||
Response 1: Thank you for your comment. While studies on TERT expression in melanomas and nevi have indeed been published, none have specifically focused on thin melanomas. This is the novelty of our study: we explore the potential of TERT immunohistochemistry to distinguish between thin melanomas and melanocytic nevi, which can occasionally pose a diagnostic challenge for pathologists. We believe that any new antibody demonstrating clinical utility can contribute to improving the differentiation of such difficult cases. |
||
Comments 2: The manuscript fails to provide a compelling rationale for studying a Romanian population, with no clear justification for potential genetic, epidemiological, or clinical differences that warrant this specific study. |
||
Response 2: Thank you for your thoughtful comment. You are right—we did not address potential genetic, epidemiological, or clinical differences, and we apologize for this omission. Our study focused on a Romanian population, as we work in a Romanian hospital and this is the patient group available to us. However, we acknowledge that future research involving multicenter populations and larger cohorts is necessary to validate our findings. Regarding genetic differences, the prevalence of BRAF mutations in Romanian patients has been reported to be similar to that in other countries (https://doi.org/10.3390/medicina60030351). Likewise, the epidemiological and clinicopathological characteristics of our cohort align with those observed in other European countries. For instance, the higher incidence of melanomas in female patients is comparable to findings from Spanish and French populations, and the median age at diagnosis is also similar. Additionally, the most common melanoma subtype in our study was superficial spreading melanoma, consistent with data from Spain and Serbia (https://doi.org/10.3390/jcm14030946). We have incorporated these points into the Discussion section [page 12, paragraph 3, lines 370-379].
Comments 3: The authors do not adequately address the well-known challenges related to antibody specificity, staining protocols, and interpretation variability in TERT immunohistochemistry, raising significant concerns about the reproducibility and reliability of their results. Response 3: Indeed, the wide variety of TERT antibodies and the conflicting results reported in the literature can be misleading. However, when focusing on studies that used the same antibody, the findings tend to be more consistent. A key factor contributing to these discrepancies is the use of different antibodies and variations in how the staining is interpreted—for example, differences in staining localization or whether the pattern is homogeneous or heterogeneous. Nevertheless, when we consider studies on other types of cancer, some commonalities emerge, such as the cytoplasmic localisation of staining often being associated with malignant tumours [page 12, paragraph 3, lines 368-370; page 11, paragraph 2, lines 306-311].
Comments 4:The limited sample size, particularly within the specific Romanian population studied, compromises the statistical power and generalizability of the findings. Response 4: Thank you for your comment. We agree that our cohort is small; however, it specifically includes only thin melanomas, which are relatively uncommon. Despite the limited size, the Romanian population is, in most respects, comparable to other European populations, allowing for the potential generalization of our findings.
Comments 5: The study's findings could have been significantly strengthened by the inclusion of molecular data, such as RT-PCR for TERT mRNA expression and/or sequencing for TERT promoter mutations, to validate the immunohistochemical results. Response 5: Thank you for emphasizing this point; we completely agree. Integrating more complex molecular data would indeed have been valuable. However, immunohistochemistry remains the most accessible and widely used method in routine clinical practice within histopathological laboratories. It plays a key role in daily diagnostics, and there is a continued need for new antibodies to support the differential diagnosis of melanocytic tumours. Previous studies have shown a low correlation between TERT promoter mutations and TERT expression, likely due to alternative mechanisms such as TERT gene amplifications contributing to its expression [page 3, paragraph 1, lines 92-93; page 12, paragraph 2, lines 361-363].
Comments 6: The study's reliability is further called into question by the absence of an evaluation of interobserver variability in the interpretation of the immunohistochemistry results, a crucial factor in such studies. Response 6: Thank you for your comment. The staining was evaluated by a single, experienced, board-certified anatomopathologist, which we acknowledge could be a limitation of the study. We agree with your observation.
|
||
4. Additional clarifications |
||
The limited number of published studies show that TERT expression detected by immunohistochemistry is highly variable. Therefore, we believe this research is necessary to help clarify and advance understanding in this area. |
Reviewer 3 Report
Comments and Suggestions for Authors
I have reviewed the manuscript titled "TERT Immunohistochemistry in Thin Melanomas Compared to Melanocytic Nevi", which presents an immunohistochemical analysis of TERT protein expression in thin melanomas versus melanocytic nevi. The study addresses an important diagnostic challenge and adds value to the limited literature specifically focusing on thin melanomas. The research design is appropriate, and the findings are clearly presented and statistically analyzed. However, the manuscript would benefit from clarification of certain methodological points, improved structure in the discussion, and stronger emphasis on the clinical implications and limitations. Specific comments are provided below.
Abstract
1. In the abstract, briefly mention the sample size and the most significant statistical finding to emphasize study robustness.
Introduction
1. Consider trimming redundant background information to enhance focus.
2. Include a more direct statement of the hypothesis and research objective at the end of the introduction.
Material and methods
1. Explain why the cutoff of >50% stained cells was chosen as the diagnostic threshold—was this based on previous studies or ROC analysis?
2.Clarify how observer variability was addressed (e.g., was scoring done by more than one pathologist? Any interobserver agreement assessed?).
3.Include a rationale for using cytoplasmic vs. nuclear localization as a diagnostic criterion, given the diversity of TERT localization reported in the literature.
Results
1. Consider presenting a ROC curve or diagnostic performance analysis to better quantify the utility of TERT IHC (sensitivity, specificity, PPV, NPV are briefly mentioned but not graphically shown).
2. Highlight the clinical relevance of differences in TERT staining in a separate short paragraph, especially for early diagnosis or pathology workflow.
Discussion
1.The discussion is too lengthy and repetitive in parts; condense overlapping findings and focus on your unique contributions.
2. Be cautious about overstating diagnostic utility. The specificity (58.97%) is relatively low, so conclusions about potential diagnostic use should be appropriately qualified.
3. Expand on why some nevi showed nuclear TERT staining while melanomas did not—is this due to antibody specificity, cell type differences, or artifact?
Conclusion
1. Briefly restate that this method may aid but not replace conventional diagnostic criteria.
2. Emphasize the need for larger multicenter studies or validation cohorts.
Author Response
1. Summary |
|
|
|
Thank you very much for taking the time to review this manuscript and for the opportunity to improve it. Please find the detailed responses below and the corresponding revisions highlighted in the resubmitted file. |
|||
|
|
||
|
|
||
2. Point-by-point response to Comments and Suggestions for Authors |
|||
Comments 1: Abstract: 1. In the abstract, briefly mention the sample size and the most significant statistical finding to emphasize study robustness. |
|||
Response 1: Thank you for pointing this out. We agree with your comment and have added the information you mentioned to the Abstract section [page 1, paragraphs 2 and 3].
|
|||
Comments 2: Introduction 1. Consider trimming redundant background information to enhance focus. |
|||
Response 2: We agree with your comment and have removed some unnecessary information from the Introduction section. Thank you again for your valuable feedback [pages 2–3].
Comments 3: 2. Include a more direct statement of the hypothesis and research objective at the end of the introduction. Response 3: Thank you for your comment. I have revised the manuscript accordingly and worked to improve the clarity of the hypothesis and research objectives [page 3, paragraph 2, lines 94–99].
Comments 4: Material and methods 1. Explain why the cutoff of >50% stained cells was chosen as the diagnostic threshold—was this based on previous studies or ROC analysis? Response 4: The cutoff was chosen hypothetically, as we were unable to perform a ROC analysis due to the data being categorized into percentage ranges rather than continuous numeric values.
Comments 5: 2.Clarify how observer variability was addressed (e.g., was scoring done by more than one pathologist? Any interobserver agreement assessed?). Response 5: All cases were interpreted by a single board-certified anatomopathologist (B.P.), and therefore, interobserver variability was not assessed.
Comments 6: 3.Include a rationale for using cytoplasmic vs. nuclear localization as a diagnostic criterion, given the diversity of TERT localization reported in the literature. Response 6: Thank you for your comment. We fully agree and have accordingly added the rationale for adopting this classification [page 3, paragraph 5, lines 129–133].
Comments 7: Results 1. Consider presenting a ROC curve or diagnostic performance analysis to better quantify the utility of TERT IHC (sensitivity, specificity, PPV, NPV are briefly mentioned but not graphically shown). Response 7: Thank you for your comment. We agree that a ROC curve would have been appropriate. However, the cutoff value was selected hypothetically, as our data is grouped into percentage ranges rather than continuous numeric values, which prevents us from performing a proper ROC analysis. If it were possible, we would have been glad to include it.
Comments 8: 2. Highlight the clinical relevance of differences in TERT staining in a separate short paragraph, especially for early diagnosis or pathology workflow. Response 8: Thank you for pointing this out. We have added a new paragraph to emphasize this point [page 5, paragraph 3].
Comments 9:The discussion is too lengthy and repetitive in parts; condense overlapping findings and focus on your unique contributions. Response 9: Thank you for your comment. We agree with your suggestion and have made the necessary modifications, removing the unnecessary information[pages 10-13].
Comments 10: 2. Be cautious about overstating diagnostic utility. The specificity (58.97%) is relatively low, so conclusions about potential diagnostic use should be appropriately qualified. Response 10: Thank you for pointing this out. You are right, and we have included additional discussion on this topic in the Discussion section [page 12, paragraph 4, lines 383–386].
Comments 11: 3. Expand on why some nevi showed nuclear TERT staining while melanomas did not—is this due to antibody specificity, cell type differences, or artifact? Response 11: Thank you for your comment. We agree with you and have added some possible explanations [page 11, paragraph 2, lines 306–314].
Comments 12: Conclusion 1. Briefly restate that this method may aid but not replace conventional diagnostic criteria. Response 12: Thank you for your comment. We have added a statement highlighting this point [page 13, paragraph 1, lines 404–405].
Comments 13:2. Emphasize the need for larger multicenter studies or validation cohorts. Response 13: Thank you for your suggestion. We have included information on this topic in the Conclusion section [page 13, paragraph 1, lines 403–404].
|
Round 2
Reviewer 2 Report
Comments and Suggestions for Authors
The comments have been addressed and the necessary revisions have been to the manuscript.
Reviewer 3 Report
Comments and Suggestions for Authors
I have no comments in this stage.